# The Fascists Are Coming! Teacher Education for When Right-Wing Activism Micro-Governs Classroom Practice

Peter Appelbaum

School of Education, Arcadia University, Glenside, PA 19038, USA; appelbap@arcadia.edu

**Abstract:** U.S. educational reform is often the harbinger of global demands on mathematics education practices globally. It behooves teacher education to 'catch up' on current trends, hopefully, to stave off the worst of the fascist tendencies of contemporary politics of education. Past foci on research-based 'best practices' and 'mathematics for all', grounded in liberal multiculturalism (confirming expectations from critical mathematics education scholarship), have become the targets of activists and politicians, turning once-exemplary teachers and their students into casualties. The four phases of *currere* are employed to study this phenomenon and to identify strategies and tactics for teacher education programs. The *currere* methodology indicates that the content of such programs must reduce time devoted to evidence and research-based practice in order to accommodate techniques and knowledge bases for the recognition of right-wing tactics, clowning, slogan parody, and political organizing. Teacher education must further place mathematics teachers' embrace of expertise, authority, and neutrality within broader perspectives on the politics of education, organizational infrastructure strategies and tactics, resource curation, and personal safety planning. Teacher educators themselves must prepare responses to threats on their careers, lives, and families, and proactive 'game plans' for the development of new program curricula.

**Keywords:** mathematics teacher education; critical mathematics education; socio-political issues; social justice

## 1. Introduction

Mathematics education seems to have reached a consensus around the commitment to creating the best experiences with mathematics that are possible for the greatest number of people. Although a range of social, political, cultural, ideological, geographic, and other perspectives are represented in approaches to mathematics teacher education globally, professional associations share a general vision of successfully reaching as many learners as possible [1–5], with related recommendations for teacher education to support their goals [6–8], as well as for those who supervise mathematics teachers [9]. This is often framed within a rhetoric of "mathematics for all" [3,10,11]. A commitment to success and increasingly sustained engagement beyond some level of "basic knowledge" is typically understood as supporting mathematics education for democracies through an educated—that is, informed and enlightened—citizenry [11–13], in which all learners, especially those who are members of socially marginalized communities, deserve the best opportunities. Alternatively, such attention to 'all' is understood as providing knowledge and skills essential to economic progress and technical or industrial innovation [14–16]. Research on effective teachers of underrepresented groups in mathematics has recognized important instructional methods based on challenging the status quo and accepting the political nature of one's professional work [17,18]. Contemporary mathematics education theory has begun the spadework of explicitly connecting instruction and curriculum development to political philosophy [19,20]. Prospective teachers are often helped to identify how to garner the support of potential gatekeepers as they pursue reflective practice [20,21].

However, political movements worldwide have created the opening for direct attacks on forms of mathematics instruction considered to be in conflict with their beliefs by bringing conservative right-wing ideologies out of previously social margins into mainstream governance [22,23]. Most of the sources mentioned in this paper date from before this huge worldwide social movement, during the period in which teacher education practice was grounded in evidence from research amassed over decades. Across the wide range of existing motivations and goals for mathematics education, a general professional agreement coalesced internationally around assessment and instruction that facilitate meaningful mathematics learning, usually described as helping learners to understand mathematical concepts and to build skills upon a conceptual framework, providing opportunities to practice problem solving with mathematics in situations that are intended to model the real world or everyday life, and taking advantage of the funds of knowledge that students bring to the school experience from their home, family, and community life. To truly believe in "mathematics for all" is to commit personally and professionally to a shared project among mathematics educators. Esteemed researchers, award-winning teachers, and highly praised curriculum materials pursued a loosely agreed-upon commitment to this version of mathematics education.

Within the fascist-aligned activism that is emerging, however, those previous authorities have been negatively branded as "woke", "teaching non-math during math", and more. "Woke" originated in African American vernacular to communicate alertness to prejudice and discrimination. In its original sense, a "woke" mathematics teacher would be aware of, and actively attentive to, important societal facts and issues, especially issues of race and social justice. Right-wing politicians and activists subsequently seized this term as a focus for fomenting fear and a way to rally public sentiment around its derisive ridicule. Mathematics standards and principles grounded in cognitive psychology, sociological understandings with decades of documentation, and data on increasing demonstration of mathematical competencies are now interpreted as "woke" and a serious threat to the public good. They emerged to support a view of mathematics for all that recognized cultural differences as relevant to school learning, the value of connecting mathematics concepts to students' everyday lives, the usefulness of problem-solving contexts, and critical thinking. They tended to be embedded in a generally neutral sense of multiculturalism and democratic citizenship. Such strategies and methods might provide a scaffold for the more political action required to survive the direct assault of fascist activists.

However, a nuanced distinction is that mathematics in school can no longer be merely used to educate active citizens who understand the rhetorical uses of mathematics in social and political life. While even informed civics is threatening to those who mean to manipulate, the recognition that the mathematics itself is embedded with the vestiges of colonialism means that our efforts are more than simple social justice. Politicians and the media that feed off them have fueled parent groups, ultra-right-wing militia groups, and some disgruntled teachers themselves, so that mainstream mathematics education recommendations are perceived as intermingled with threats to white supremacy and other conservative values. Because of this, the culture wars of mathematics need to be a central component of teacher education. What used to be the recruitment of gatekeepers to understand how meaningful mathematics practices can facilitate the gatekeepers' goals [21] has turned into the need to protect oneself and one's family from physical harm, rather than to solicit approval or direct forms of support. While a school administrator once would mediate family complaints about problem-based learning, for example, is now more often a short office visit to terminate employment and escort the teacher from the school. During a less overtly political time in recent history, a teacher would justify research-based practices with multicultural explanations and performance assessment data. Now, teachers have to decide whether they can live with the internal conflict between knowing what the best practices demand and the long-term harmful effects on reduced learning and participation in mathematics that are the known outcomes of the curriculum they are required to implement. Intersecting with such issues is the need for teacher educators to

practice self-care and professional development, no longer relying on the assumption that their professional positions convey status, authority, immunity from danger, or sympathy for their personal fears.

All of this might seem like a predictable outcome of neoliberal policies and practices. Critical mathematics education scholars have been describing for decades how the constellation of corporate interests, liberal politics, neoliberal global economic interests and transnational competitions for status and its privileges create supposed "crises in mathematics achievement" buttressed by industries that "measure" achievement, industries that train and "teacher-proof" practices, and industries of scholarship that position academic researchers and practitioners, along with students and families as pawns in a perpetual coalescence of profit-driven needs construction [24–27]. A key point of my analysis to follow is that we need to move past critiques of the neoliberal–transnational nexus of power and act, now and without delay within all positions of power and knowledge, to a fundamental dismantling of mathematics education in all forms.

This article outlines an approach to teacher education that prepares teachers, and those who train and support them, for the political realities of our time. I focus heavily on the situation in the United States. This is a reasonable choice since many innovations and resources originate in the United States. Trends and fads often start in the United States and spread later around the world. The main idea is that U.S. educational reform is often the harbinger of global demands on mathematics education practices globally. It behooves teacher educators to "catch up" on current trends, hopefully to stave off the worst of the fascist tendencies of contemporary U.S. politics of education as it spreads worldwide.

## 2. *Currere* Methodology Applied to the Circumstances

The methodology guiding this research report is an analysis of news reports and other artifacts of scholarship guided by the four-phase *currere* method of inquiry [28]: (1) regressive: begin with an autobiographical inquiry; (2) progressive: turn toward the imagined future implications of current experience; (3) analytical: interpret the autoethnographic past, present and future; and (4) synthetic: use fragments of experience and artifacts within the other phases to understand the larger cultural and political context, and to make action decisions consistent with one's values, commitments, hopes and dreams. *Currere* work does not necessarily follow a linear sequence from one phase to the next. Some of the work from each phase seeps into the others as the analysis takes on a life of its own, often leading to questions, rather than starting with them, generating hypotheses within the work rather than as starting points for data collection. Scholars experienced in *currere* enter the experience through autobiography, not intending to write memoirs or life histories, but rather to tap into the artifacts of everyday life shared by culture and society. The artifacts do not need to be carefully selected, since any artifact is an opening into the themes, patterns, ideologies and relationships of power that become evident through the *currere* processes [24,25]. The four phases ideally lead to the sort of richer understanding of self-transformations necessary for work toward social justice [26]. As discussed below, teacher educators' reliance on the authority of science (evidence-based research and classroom practices) has mostly amplified the very problems inherent in the politicization of mathematics teaching and learning. If we view teacher education and the self-study of teacher education as disciplines in their own right, then alternative methodologies and perspectives such as phases of *currere* are ways to "stretch [them] from the inside to provide richer, more meaningful studies" [27], p. 290. Teacher education and teacher educator self-study are like mutually concentric circles of perspective sharing porous boundaries of relation among knowledge, institutions, expertise, authority, justice, and community. Both also share a particular relationship with time and space, in the sense that teaching and teacher education are always at once about the present, the past, and the future, with what happens in the moment always related to past experiences and imagined futures. The scholarship in this article is grounded in *currere* methods since they have the potential to focus attention on such "borders between what is and what is

becoming, and between notions of the particular and the whole ... often configured in terms of interpenetrating circles, or spheres of temporary coherence" [29], p. 93.

### 2.1. Regressive Phase of Research

In the mid-1990s, a year or so before my university was to make its decision about my tenure, my provost received a thick dossier documenting the ways that my scholarship was dangerous to all humankind. I was a strong advocate for new reforms in mathematics education grounded in problem solving, reasoning, communication, multiple representations, writing and literature in mathematics, and interdisciplinary problem-posing approaches. Conservative mathematicians and members of the general public sympathetic to their cause compiled this dossier because they believed that traditional "drill and practice" on basic skills should be the sole experience for all mathematics learners. I was part of a program of "citizen resistance" to the changes supported by mainstream mathematics education research and professional associations, which was slowly influencing policy and practice. In my own case, I was lucky at my fairly liberal, Northeastern U.S. State University. My provost forwarded the dossier to me, with a post-it note saying, "Congratulations on being at the center of controversy in your field." It is possible that this dossier ironically helped me to obtain tenure.

That frightening experience has many precedents in the history of (mathematics) education and has turned out to be a foreshadowing of far more terrifying and dangerous actions that have been percolating—and now taking visceral, real-life forms—in the current socio-political climate. To take one current example, I was horrified in March of 2023 to read of the eminent and esteemed Stanford University mathematics education scholar Jo Boaler, who posted an update on her blog, recounting the personal threats to herself and her family that have been mounting in specificity over the past few years [30]. I experienced an intense bout of PTSD. In her blog post, Professor Boaler supports serious and aggressive debate within a democracy: "Honest academic debate lies at the core of good scholarship." Yet, she asks, "What happens when, under the guise of academic freedom, a small cluster of aligned people distort the truth in order to discredit someone's evidence and boost their own allied position?" She had listed details about personal and academic attacks on her blog since 2012. Yet, these attacks had now escalated. Noting connections between the "disagreements" over research-based practices supporting equity and mathematics for all and her opponents' histories of racist comments, she felt compelled to share the escalation of their attacks, not only upon her, but now upon her family and the members and families of the other four authors of a newly proposed mathematics curriculum framework for the state of California that included realistic threats of physical violence. Approximately one-third of California students demonstrate proficiency in mathematics [31]. Data on high levels of mathematics show indefensible social and racial inequities [32]. Boaler's research has persistently demonstrated the efficacy of active engagement in mathematics learning in comparison with simply practicing procedures. This is consistent with an enormous body of supportive literature from other scholars [33]. Echoing my earlier experience with these folks in the 1990s, yet at a much more sophisticated and heightened level, the most recent and intense time of doxing and online harassment of Boaler included the sharing of her personal emails and very personal details of her daily life on Twitter, and rallying supporters on the Fox News Channel's Tucker Carlson program, leading to the reported direct threats from the many Twitter followers of the opponents and Carlson viewers. Repeated additions of slurs and attacks on her Wikipedia page led Wikipedia to delete them and lock the page. Stanford University police have determined it necessary to include her home in their daily patrols to ensure the safety of her and her family.

### 2.2. Progressive Phase of Research

Boaler's story is just one small piece of a larger social movement discrediting mainstream professional standards in mathematics education as connected to other politically charged yet research-based reforms in education. The state of Florida recently banned

several highly regarded textbook series produced by well-established publishers [34]. Local school districts nationwide are following suit with similar restrictions on what teachers can and cannot do in their classrooms. Teachers in my graduate program courses at my own university tell tales of secretly reading a poem at the start of a lesson to motivate students, fearful that an administrator casually walking by their classroom might overhear and punish them for "not following the script." Others are required to attend so-called "professional development" workshops that "train them" on appropriate algorithms to demonstrate while recognizing "woke" mathematics to avoid in the teacher guides that accompany their text materials. *The New Yorker* magazine, usually devoted to literary essays, arts, and cultural commentary, found this movement so noteworthy that they published a lengthy, historically grounded feature article [35], tracing long-standing controversies to the ways that mathematics is often imprisoned by its associations with order and discipline, through the Sputnik era that sparked a democratization of mathematics beyond mere skills and the 1970s "Back-to-Basics" backlash, the 1980s NCTM unleash of "problem solving", the 1990s backlash labeling problem solving "fuzzy math", and finally the current versions of "social justice" that struggle to link school mathematics with life opportunities beyond low-level skills.

Mathematics educators have always understood the ways that their curriculum can support or challenge contemporary political movements [36]. Textbook writers as diverse as David Eugene Smith [37], World War II German text writers [38], and contemporary writers of mathematics textbooks for social justice [39–41] are excellent examples of what is possible. The unfortunate aspects of the current politicization of mathematics take their most frightening form in the personal attacks on teachers and curriculum developers. Nevertheless, the more pressing concerns have to do with the ways that these attacks are physical reductions of a misunderstanding of research-based practices. As Brian Lindaman, Chair of the California Framework Committee, is quoted in *The New Yorker*, it is indeed important to ask teachers to take on the challenge of addressing equity by finding ways to help all "students find the joy and beauty of math early" and "Many of them do, it's just that somewhere along the line that gets taken out and they stop seeing the beauty in mathematics. And we think that has something to do with the way it gets taught." Yet, the opponents of research-based practices have reduced all of the proven effective pedagogical techniques into a simplistic misrepresentation of research's support for "the importance of children seeing real-world applications of what they were doing, in a way that made sense to them" [34] with a naïve misconstrued misconception of "wokeness". Since wokeness is linked to every malady from mass shootings to lower military recruitment, inflation, youth unemployment, and more, the public micro-management of classroom practices from outside the classroom demonstrates a social fear of science-based policy. Boaler directs a research center at Stanford that works jointly with neuroscientists on brain development and brain-based classroom practices. The movements' rigid adherence to mathematics as purely utilitarian skills rather than a meaningful collection of concepts through which life can be enriched also limits the potential for creative lessons. In the Pennsylvania town of Perkasie, the school district felt it necessary to apologize to parents and students after one of its high school teachers assigned mathematics homework that included what the district is calling "adult content without a proper context" [42]. The assignment focused on the highly regarded autobiography, *I Know Why the Caged Bird Sings*, by the acclaimed and celebrated poet Maya Angelou. Despite Angelou's stature, one parent declared, "I still think it is a problem . . . It shouldn't be on a math test" [42].

### 2.3. Analytical Phase of Research and Tentative Conclusions

One type of response taken by mathematics educators to this public disparagement of their work is to dig further into the research literature that supports their recommended practices. Boaler itemizes the attacks on her work to be specifically about Research on Timed Testing and Math Anxiety, Research on Mindset, and Neuroscientists' Studies of Brain Responses to Mistakes [43]. This strategy is highly ineffective as a reaction to

the manufactured controversies, since the threatening opponents dismiss any academic, research-based evidence as further examples of what they oppose. The anti-intellectualism of this dismissal is historically linked to fascism and terrorism [44]. A country that is not officially fascist in its doctrines can experience fascist politics. Those politicians base their efforts on dividing society and demonizing specific groups. Anti-intellectualism is one of the "pillars of fascist politics", along with myths of certain subgroups as superior to others. The micro-management of mathematics education practices is one example of the overarching social tensions plaguing the United States and other countries around the world today. Stanley [44] describes the intimidation of scholars into silence as one key feature of fascist politics. Gender studies scholars, scholars of Islam and the Middle East, and those, for example, in African American Studies and Indigenous Studies whose work threatens a mythic, rosy picture of the past they wish our education systems to present as objects of veneration, are equally attacked alongside mathematics education researchers and policy innovators. By harshly attacking those who seek to show the truth in its full complexity, writes Stanley, the activists threatening them undermine the search for truth and valid, academically informed professional practices. These overarching social tensions come full circle in establishing right-wing activist methods, platforms, and discourse that are readily applied to school mathematics in addition to other disciplines and general school policies. At the same time, popular views of mathematics as utilitarian skills make it possible for engaging in meaningful mathematics practices to be experienced as threatening to the false security of rules and memorized, "correct" ways to behave. Options and nuances in mathematics, when they accompany flexible inclusive and culturally welcoming school policies, seem like the ground is shifting under one's feet.

Much as we mathematics educators want to focus on known best practices for classroom teaching and learning, the implications of the current political trends are that we must simultaneously gain the skills and practices to fight the fascist takeover of our political institutions as well as the media coverage of our classroom practices. Right-wing extremists have two main strategies. Prospective teachers need practice in recognizing and responding to these strategies.

One is to set up situations where they can play the victim and increase sympathetic interest in their cause, or at least polarize and confuse the issues. Their other favorite tactic is to threaten and use violence to increase the fear level of their opponents. Symbols are less costly than actually injuring and killing, and so they like to use symbols like clubs, tiki torches, burning crosses, or dressing in sheets or military-style uniforms. By getting there first, they set the tone, but they do not win just by doing that. Their victory comes when their opponents respond in a like manner and try to out-intimidate the intimidators [45]. Successful anti-fascist strategies involve a commitment to re-framing the issues and confronting the right-wing escapades with a strikingly different tone that confounds them [45–47]. Boaler's attempts to calmly list the substantial evidence for her work set a different tone. She maintains a stance of expertise and cultivated neutrality, clothed in the historically rhetorical use of science and the scientific method grounded in data as defining truth. This fails to counter the fascists because they begin with a different rhetoric; the fascists begin with faith in certain beliefs, and use their faith to determine their own truth, distinct from data that they have no confidence in. Rochelle Gutiérrez [48], in contrast, leaps directly to the motivations behind, and social consequences of, traditional skill-based mathematics education practices: "The majority of the messages I received (A sample from Gutiérrez' article: 'Whites will always rule and always achieve . . . I suggest you go and shoot yourself . . . Someone as evil, stupid and racist as you should be barred from teaching . . . I'm going to spearhead a media campaign to have you fired.' [48], p. 72) were racist, misogynistic, and vulgar. Responders did not seek dialogue, analyze the argument I made, or even maintain a context of mathematics education. Instead, the messages were an attack on the person, a smear campaign" [48], p. 70. A case study of Boaler's experience should be a central feature of teacher education in mathematics so

that future teachers can understand why explanations grounded in research are not well received and serve only to fuel the flames of right-wing activists.

Other professionally informed mathematics educators use the strategy of clowning—resisting authorities not by direct resistance or arguments, but by participating in ways that make it clear how those in power are ridiculous and silly. For example, they might show up in enormous numbers at every public event, such as school board meetings and government policy debates, in absurd costumes that mock the idiocy of the right-wing positions. Clowning puts the fascists in the awkward position of not being able to claim victimhood or a need for aggressive behaviors. Stealing the empty fascist slogans and replacing them with highly similar but silly alternatives repeated loudly often defuses the originals. Or, mathematics educators might satirize expectations or policies in events with exaggerated parody. Imitations of what right-wing politicians and school administrators demand can be performed in public places in ways that make the outrageously harmful or demotivating forms of classroom practice visible. Memorial events for the victims of right-wing actions (horrible test scores, loss of job prospects, lower incomes, increased health risks, and so on) create public displays counter to the fascist messages. Establishing safe houses to repair the damage wrought by fascist extremism (community-run mathematics circles using state-of-the-art practices) can protect the defenseless victims of fascist policies while telegraphing to the wider public how similar the expected practices are with more overt forms of abuse that require "safe houses". Prospective teachers can practice such actions. They can gather with activist teacher groups at public venues to clown, create posters and protest signs for school board and local government meetings about school policies and programs, and invent their own satirical events that highlight the dangers of bad practices that are finding increasing prominence. Teacher educators and their protegees might study the Orange Alternative activists in 1980s Poland [49,50] for inspiration and examples of humorous tactics that defused personal risks of violence at the same time as communicating a deeper and more substantial critique than would have been possible with direct evidential confrontation. The Orange activists dressed as silly clowns and gave the police who were intending to perform crowd control flowers and kisses; they placed small gnomes all over the city without explanation overnight, to gently use non-threatening decorations as a sign that dissenters were everywhere, ready to sprout and be seen at a moment's notice. Experiential curriculum using Theater of the Oppressed techniques for working through alternative options for classroom practice has been highly successful in teacher education [51,52]; teacher educator self-study groups, coursework in teacher preparation, and professional development workshops can use the same techniques to role play, problem solve, and imagine action options in proactively anticipating or responding to right-wing hostilities and policy development.

Mathematics educators have 'grown up' in an ideology that believes in the efficacy of critical thinking. Our own successes with mathematics have often led to admiration for what we think of as our intelligence. We often consider a good mathematics student to be "better than just a good student". We are comfortable with a position of power and privilege tied to the assumption that mathematics is an essential school subject. We take pride in that presumed necessity. Since few people are as knowledgeable about mathematics as we are, we assume that we should be able to define our work and judge how well we are at it. Most of us believe that mathematics is neutral, that it is people applying mathematics that create its political uses, rather than imagine that there are ideological or political bases to our mathematics. Ethnomathematics, critical mathematics education, and social justice mathematics each challenge that naïve standpoint. Yet, it is not simple even for us to interrogate the underlying ideology of our discipline. The complex history of school mathematics, with its European origins, overlaid upon a global and transnational variety of national school systems, with their own histories of colonialism, Indigenous communities, social movements, and cultural and political conflicts, is challenging and often overwhelming. In the current climate, scholarship and social movements describing and challenging the oppression that many have suffered (post-colonial nationalism, queer

rights, women's rights, Indigenous rights, etc.) become direct threats in the zero-sum culture war wielded by aggressive, right-wing activists. School mathematics is the ultimate example of how the legacies of colonialism are entrenched ideology, since it is usually believed to be universal and neutral, despite its very specific history. Teaching traditional mathematics curricula perpetuates the epistemicide and erasure of other mathematical traditions and cultural practices worldwide [53]. Ethnomathematical attempts to integrate local mathematics practices with traditional school mathematics often perpetuate coloniality by only understanding local practices in terms of the traditional curriculum, rather than using local practices to reconceptualize what mathematics is in the first place [54,55]. To act on these realities really would change the nature of school mathematics! Informed mathematics educators would seem to be ethically obligated to act on this understanding, that is, to accept the threatening nature of what they are doing when they make it possible for school mathematics to be meaningfully learned by all students. In other words, what we believe in is, we need to admit, threatening to fascists. The fascists know this better than we do [36].

It is time to double down on the important collective work of clowning and safe houses, and zero in on the re-framing that we otherwise would not think we should need to do. We cannot merely claim our expertise and expect folks to accept our pronouncements about best practices. We must mock the opponents, and protect each other from their personal violent attacks. They are coming. Believe me. I know. What skills must we teach teachers, as they come under attack? Gutiérrez [46] suggests the following: that political knowledge is equally important to cognitive psychology and subject matter content; how to find curricular materials that challenge dominant paradigms of mathematics as a neutral collection of concepts and skills; organizational infrastructure strategies and tactics (tools and techniques of documenting attacks, countering them, and protecting colleagues personally and professionally); and resource curation (networks, public media contacts, sympathetic policy wonks, etc.). This is not a huge leap from some of what we already do. Yet, it demands that we think more about how to shift from trying to convince teachers with little to no experience as learners in model mathematics classrooms to use "best practices"—introducing multiple representations, building understanding through classroom conversation and students' listening to each other, and integrateing mathematics with the arts, reading, and writing. This shift requires apprenticeship beyond an embrace of new instructional methods in the political action and defensive tactics required to survive while following those recommendations for best practices. Ladson-Billings [17] could gently talk about teachers seeing themselves as political beings. Now that is not enough. One must actually act politically just to be able to do the most highly recommended work.

## 3. Synthetic Phase and Results

The first three phases of *currere* took us on a journey from personal autobiography into a process of extrapolation and alarming trajectories, demonstrating the personal dangers that mathematics teachers face in the midst of right-wing conservatism. These personal dangers reflect dismal demands that the teachers implement less than optimal instructional strategies, combined with potentially harmful forms of assessment and evaluation. The consequences of such micro-governance of classroom curriculum and instruction include knowledge gaps, or ignorance, regarding self-care and evidence-based best practices and a naïve adherence to an ideology of expertise, which promotes ineffectual resistance. These consequences point to the need for techniques of self-care and political action training for both teacher educators and their students.

### 3.1. Knowledge Gaps

Through questions about the historical context that guided a genealogy of the circumstances surrounding contemporary teacher education, it became evident that liberal commitments to diversity and "mathematics for all" are inadequate as principles or standards for mathematics education. Teacher education needs to reconsider its positionality in

terms of social justice, supporting high achievement for all students, and training teachers in evidence-based practice. It is not that evidence-based practice is bad. What is problematic is how the attention to practicing professionally recommended instructional methods creates knowledge gaps. Both teacher educators and the new teachers (indeed, most mathematics teachers) presume an authority originating in their subject-related and pedagogical expertise. This presumption does not prepare them for criticism, nor for threats on their lives and the lives of their family members. Critical teacher education has tended to emphasize the use of evidence-based and research-based justifications for instructional innovations [56]. This emphasis leaves teacher educators and teachers ignorant of the strategies they need to protect themselves and to provide students with a socially just mathematics education, as well as the ways that seemingly good commitments merely sustain and perpetuate forms of inequity tied to the legacies of colonialism [53].

### 3.2. Critical Mathematics Teacher Education Leaves 'Believers' Unprepared

Programs that encourage such an orientation to mathematics teaching hope to produce mathematics teachers committed to "mathematics for all" and other socially responsible goals tied to social justice and environmental stewardship [57,58]. Such a perspective on mathematics teacher education buttresses the ideology of expertise, ironically feeding into the perception by right-wing activists that mathematics teachers are misguided crusaders for "social justice" and, thus, in need of strong motivations to change their practices. The perspective further leaves teachers unprepared for hostile takeovers of the curriculum and school policies, because it does not think of political readiness as an important component of teacher training. For example, Skovsmose [59] clarifies critical mathematics education pedagogy requires two criteria, subjective and objective. Teacher educators use these criteria to provide prospective teachers with techniques for selecting appropriate mathematics problems [60]. In other words, critical mathematics education and other versions of mathematics for social justice help teachers to further believe they are highly trained and knowledgeable, and that the knowledge that matters most is the knowledge that helps them to teach specific content through well-chosen content-based activities. The subjective criterion requires that the problem appear relevant to students within their conceptual understanding. The objective criterion requires the use of data and detail to view an existing social issue in order to facilitate deeper understanding. The integration of mathematics and social justice is claimed to potentially spark meaningful conversations about issues impacting local communities and beyond, preparing individuals for citizenship. Neither criterion helps when conceptual understanding, the integration of social justice values, and active citizenship are both ridiculed and feared by members of the community.

Critical mathematics education as an orientation promoting critical citizenship and democracy [60–62] might have circumvented the current situation. The present moment requires something other than a critique, a style of coalition building across the critical democracy–liberal divide that recognizes a potential destruction of school mathematics altogether, with teachers and students as the primary causalities, and researchers as bystanders. My argument is to begin with the political action training I sketch in the next section, while saving the more nuanced political critique.

### 3.3. Political Action Training Can and Should Be Major Components of Teacher Education Programs

Several specific forms of training in strategies and tactics have been highlighted for responding specifically to fascist activism. These include, first of all, techniques for recognizing when curricular changes or education policies as ultra-right-wing attempts to micro-govern mathematics teaching and learning, such as setting up situations where conservative members of the community can play the victim to polarize and confuse the issues while increasing sympathy for their cause; threatening and using violence to increase fear; spreading communities with symbols that evoke the threats while not necessarily carrying them out; and intentionally using rhetoric that does not include evidence-based

research. Teacher educators and their protégées are further aided by the understanding that rhetoric of expertise and shock at the attacks on their authority grounded in knowledge are useless as tools of resistance to the threats. Successful organizing that can be practiced within teacher education includes clowning, parody, the establishment of "safe houses" and the rehearsal of responses informed by the traditions of the Theater of the Oppressed [63].

Malcolm X [64] described the problem of liberal do-gooders in 1965: they are no different from right-wing reactionaries except that they are more hypocritical. They pretend to be friends with the oppressed in order to use them as tools in an ongoing game of power with the reactionaries. Smart community organizers know that this level of self-awareness is best nurtured through political action with a common agenda [65,66]: "we need to be planning . . .—organizing ourselves, strengthening our networks, building resources and models, setting precedents, and creating infrastructure and policies . . . and professional organizations" [66], p. 20.

## 4. Discussion

Mathematics education theory development has struggled to find a place for political interpretations of practice. Certainly, there has been valuable scholarship contributing to our understanding of both the content and the orchestration of curriculum as political [67–76]. Yet, this steady accumulation of scholarship does not prepare even highly esteemed scholars of mathematics education for a political onslaught against "mathematics for all". It once seemed like our project was to, first, create models of mathematics education that recognize the diversity of the populations that they serve [77]. Tony Brown [77] suggests that "discourses of mathematics education research often aspire to cultural and historical continuity whilst simultaneously operating on the notion of a consensual ideal dependent on the future achievement of social models with adequate levels of resources". Such discourses, he argues, "rest on oversimplified models of social change that inflate the operative role of individual teachers and mathematics education researchers in affecting broader teaching and learning cultures" [77], p. 1. Ethnomathematics-informed teacher educators recognize deeply entrenched legacies of colonialism that place Tony's well-formulated argument in an even more challenging context [59]. The *currere* experience reported in this article further suggests that the nuances of ethnomathematics also simplify teacher education discussion by not attending simultaneously to the impending backlash that will accompany the successful integration of social justice with mathematics education. Political action training is "the next step".

**Funding:** This research received no external funding.

**Institutional Review Board Statement:** Not applicable.

**Informed Consent Statement:** Not applicable.

**Data Availability Statement:** Not applicable.

**Conflicts of Interest:** The author declares no conflict of interest.

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
