# Peer review of "The Fascists Are Coming! Teacher Education for When Right-Wing Activism Micro-Governs Classroom Practice"

_education, doi:10.3390/educsci13090883_

Round 1

Reviewer 1 Report

See attached document. The front end needs some work and the paragraphing throughout. But this is a strong and valuable paper that needs to be read by the field. Addressing some of the clarity concerns will improve its readability.

Excellent. Paragraphing issue - they are too long and contain multiple ideas that don't always fit together.

Author Response

I appreciate the attention given to my sometimes clumsy and long paragraphs, and used each note to find places where I can divide paragraphs. I also added the citation about the influence of the U.S. reforms on math education reforms internationally.

Reviewer 2 Report

The writer needs to reevaluate where the right-wing fascist attack on multicultural math education. Education in the U.S. and elsewhere in the world has been colonized by corporations (standardization, privatization, attacking unions, etc). The direct cooperation of neo-liberals and neo-conservatives has implemented these education policies. 

it is not clear in the manuscript what is the primary reason for inequality and unjust in math education; class or race or what? That is a crucial question... research-based approaches or best practices are slogans of neoliberal policies that professionalized teachers and destroyed public schools. in what way are they to protect teachers from fascist attacks??

Note that a liberal approach to inequality and fascist attacks on education can not solve any problems, as it is the root cause of the situation you describe in the manuscript.

Also, you need a more comprehensive literature review on critical math education. Skovsmose and others (including myself) have done work to form and theorize a math education oriented toward critical citizenship and democracy. 

Author Response

I appreciate the attention given to my manuscript, and have used each point to improve my piece.

Round 2

Reviewer 2 Report

I still think this article has great potential to contribute to the literature. However, this new version has not sufficiently responded to my previous comments/suggestions.

While neoliberal identity politics and market-driven corporate agendas in education have created material conditions for an open process of fascism, how and why should we move "past criticisms of neoliberalism"?

To improve it, the manuscript should make clear what is the definition of fascism taking place in US schools and what is its ideological-political connections&distinctions with neoliberalism and neoconservatism.

Author Response

The point of this article is to be read by even neoliberals. Critiquing neoliberalism gets us nowhere, it is actually an example of adhering to one’s supposedly superior position and status — the very failure of mainstream mathematics educators. The serious threat is now. Action is needed. Explaining how we got here can be a small piece of training in political organizing and activism, but continuing to do the critique is navel gazing. 

we have that critique- we know it well… has it prevented the right wing takeover of schools? (No)